# REALISTIC SURGICAL SIMULATION FROM MONOCULAR VIDEOS

## ABSTRACT

This paper tackles the challenge of automatically constructing realistic surgical simulation systems from readily available surgical videos. Recent efforts have successfully integrated physically grounded dynamics within 3D Gaussians to perform high-fidelity simulations in well-reconstructed static simulation environments. However, they struggle with the geometry inconsistency of simulation environments and unrealistic physical deformations of soft tissues when it comes to dynamic and complex surgical processes. In this paper, we propose SurgiSim, a novel automatic simulation system to overcome these limitations. To build a surgical simulation environment, we maintain a canonical 3D scene composed of 3D Gaussians coupled with a deformation field to represent a dynamic surgical scene. This process involves a multi-stage optimization with trajectory and anisotropic regularization, enhancing the geometry consistency of the canonical scene which serves the simulation environment. To improve the realism of physical simulations, we implement a Visco-Elastic deformation model based on the Maxwell model, effectively restoring the complex deformations of tissues. Additionally, we infer the physical parameters of tissues by minimizing the discrepancies between the input video and simulation results guided by estimated tissue motion, ensuring realistic simulation outcomes. Experiments on various surgical scenarios and interactions demonstrate SurgiSim's ability to perform realistic physics-based simulation of soft tissues among surgical procedures, showing its enormous potential for enhancing surgical training, planning, and robotic surgery systems.

## 1 INTRODUCTION

Realistic surgical simulation systems are pivotal in enhancing clinical training, offering substantial benefits for training surgeons, and advancing the development and deployment of robotic surgery systems (Huang et al., 2023; Long et al., 2023). Currently, mainstream surgical simulation systems, such as LaparoS $^{TM}$ from VIRTAMED, are commercial products that primarily utilize mesh-based technology. While these systems provide highly realistic simulations, their performance and reliability heavily depend on labor-intensive mesh modeling and manual physical parameter settings. This dependence on the skills of modelers and animators limits the variety of possible simulation scenarios. In this paper, we aim to develop a high-quality and flexible simulation system capable of automatically converting real surgical videos into surgical simulation environments as well as performing high-fidelity and physically-accurate simulations.

Building high-quality surgical simulation systems from readily available surgical videos poses significant challenges, including the reconstruction of simulation environments from monocular videos with complex rigid and non-rigid deformations and accurately modeling the soft tissues for realistic surgical simulation. Fortunately, recent advancements in scene reconstruction and physically grounded dynamics offer plausible solutions. Methods based on Neural radiance fields (NeRF) (Wang et al., 2022; Yang et al., 2023b;a) and 3D Gaussian Splatting (3DGS) (Wang et al., 2024; Yang et al., 2024b; Liu et al., 2024b) have demonstrated impressive capabilities in reconstructing surgical scenarios. Besides, PhysGaussian (Xie et al., 2024a) combines continuum mechanics with 3DGS and employs the Material Point Method (MPM) to simulate realistic motions in reconstructed models under simple point load. However, despite these technological strides, two significant challenges still obstruct the realization of the target system.

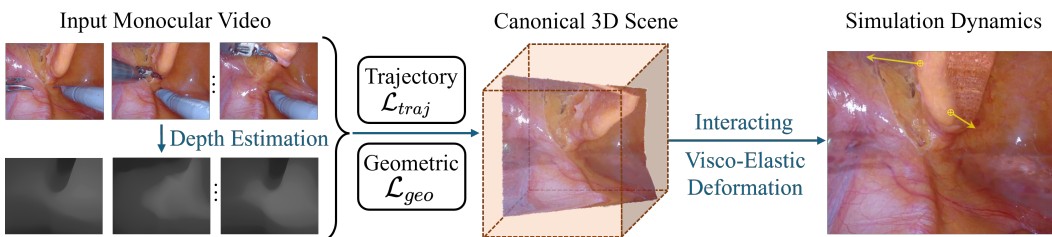

Figure 1: An overview of our method, SurgiSim. From input surgical video and estimated monocular depth, we construct a geometrically consistent canonical 3D scene through multi-stage optimization with trajectory and geometric regularization. This scene is then parameterized into a simulation environment using a Visco-Elastic deformation model, enabling realistic interactions with soft tissues and accurate simulation dynamics.

Firstly, current methods can't **reconstruct geometrically consistent simulation environments**. Recent advances in dynamic tissue reconstruction have shown promising capabilities by modeling tissues using canonical scenes with deformation fields (Xie et al., 2024b; Liu et al., 2024b; Wang et al., 2022; Yang et al., 2023b;a). However, these methods typically treat dynamic tissue reconstruction as a task of synthesizing high-fidelity images at specific timestamps. Under this definition, the deformation field is optimized to produce visually plausible results for each individual frame rather than maintaining consistent geometry across time. Consequently, the geometry mapping between the canonical scene and different timestamps becomes physically implausible. In surgical simulations, these geometric inconsistencies lead to significant artifacts such as fragmentation and messy deformations. Thus, it is crucial to develop a method capable of efficiently transforming surgical videos into high-quality, geometric, consistent, and simulation-ready environments.

Secondly, current methods fail to **simulate complex tissue deformations**. Current representative simulation methods, such as PhysGaussian (Xie et al., 2024a), primarily employ a Plastic-Elastic model with uniform simulation parameters assigned manually across the entire subject. This design falls short in capturing the complicated deformation behaviors inherent in real surgical scenarios. Moreover, the manual assignment of physical parameters to the simulation subject compromises the realism of the simulation. Recent research (Zhang et al., 2024; Liu et al., 2024a) propose to leverage diffusion priors, either in the form of generated videos or through score distillation sampling (SDS, Poole et al. (2022)) to estimate physical parameters and avoid the need for manually assigned parameters. While these diffusion-based methods are effective in simple scenarios such as handling natural fluctuation, they fail in more complex surgical situations that involve external forces from operations like pulling or cutting. Moreover, current medical video generation models (Li et al., 2024a; Sun et al., 2024) are still in the early stages of development (Cho et al., 2024), and the quality of their output is insufficient to provide accurate guidance for simulations.

To address the aforementioned challenges, we propose SurgiSim, an automated system for accurate, high-quality surgical simulation leveraging readily available surgical videos. SurgiSim incorporates a novel reconstruction module that employs 3D Gaussians (Kerbl et al., 2023), an explicit 3D representation capable of real-time rendering, to construct a canonical scene from surgical video footage. This module utilizes multi-stage optimization with trajectory and geometric regularization to enhance geometry consistency, complemented by a surface thickening method to enrich reconstructed tissue content for improved simulation fidelity. To accurately model tissue deformations across various surgical scenarios, we implement a Visco-Elastic deformation model based on the Maxwell model (Johnson & Quigley, 1992). We further employ a physics-guided parameter estimation method to acquire accurate physical parameters of tissues, leveraging observed physical effects in the input video. Specifically, we utilize point tracking (Le Moing et al., 2024) and depth estimation (Yang et al., 2024a) to derive the tissue motions induced by external forces, with physical parameters optimized through differentiable simulation and rasterization. This method ensures high-accuracy simulation with realistic physical effects, closely mimicking the complexities of real surgical scenarios. Extensive experiments across diverse surgical procedures demonstrate SurgiSim's capability to realistically simulate soft tissue interactions, highlighting its potential for enhancing surgical training, planning, and robotic surgery systems.

We summarize our contributions as follows.

1. An automated framework that enables simulations on monocular surgical videos, achieving realistic physics-based tissue deforms among various surgical scenarios and interactions.

2. A novel multi-stage optimization strategy for 3D Gaussians that ensures geometric consistency across temporal deformations, enabling reliable simulation environments.

3. A Visco-Elastic deformation model based on the Maxwell model, coupled with automated physical parameter estimation from video observations using point tracking and depth estimation, facilitates high-fidelity and accurate surgical simulations.

## 2 RELATED WORK

**Dynamic Scene Representations.** The rapidly advancing field of Neural Radiance Filed (NeRF, Mildenhall et al. (2021)) and 3DGS (Kerbl et al., 2023) has garnered significant interest for dynamic scene reconstruction. Early works based on NeRF Bansal et al. (2020); Cao & Johnson (2023); Fridovich-Keil et al. (2023); Li et al. (2022a) model scenes with implicit representations, which are difficult to interact with and therefore unsuitable for simulation. Later works based on 3DGS (Wu et al., 2024; Luiten et al., 2023; Lu et al., 2024; Li et al., 2024c; Yang et al., 2023c) use explicit representation, but require multi-view images or videos with severe movement of cameras, which is hard to obtain in the field of surgery. Recently, some other research (Wang et al., 2022; Yang et al., 2023b;a; 2024b; Liu et al., 2024b; Xie et al., 2024b; Li et al., 2024b) has focused on the reconstruction of dynamic surgical scenes. However, these studies either still use implicit representation, or primarily aim to reconstruct the dynamic scenes at specific timestamps, instead of the geometric continuity and consistency of the reconstructed canonical model.

**Material Point Method.** Material Point Method is a hybrid Lagrangian/Eulerian discretization scheme for solid mechanics (Hu et al., 2018b). The MPM system's inherent ability to handle topology changes and frictional interactions makes it well-suited for simulating a wide range of materials (Xie et al., 2024a). including elastic objects, sand, cloth, hair, snow, lava, and viscoelastic fluids De Vaucorbeil et al. (2020); Daviet & Bertails-Descoubes (2016); Fu et al. (2017); Han et al. (2019); Hu et al. (2018a); Jiang et al. (2017); Wolper et al. (2019). In addition, modern implementations of MPM utilize the parallel ability of GPUs to achieve advanced efficiency(Dong & Grabe, 2018). Recently, Xie et al. (2024a) uses GPU-based MPM to efficiently incorporate dynamics into different scenarios using a unified particle representation within the Gaussian Splatting framework. However, their methods rely on manually assigned physical parameters, and assume the parameters to be the same over large regions. Follow-up works by Zhang et al. (2024); Liu et al. (2024a) further utilize the differentiable ability of the simulation process to estimate the parameters in MPM guided by Diffusion Models (Blattmann et al., 2023). These methods extract the physical prior from generative models, which is unreliable in the surgery field.

**Surgical Simulation Systems.** Medical simulation systems are based on the deformation models used to handle tissue motion under the interaction of external forces (Meier et al., 2005). Early systems apply mainly heuristic approaches like deformable spines, spring-mass models, and linked volumes (Cover et al., 1993; Keeve et al., 1996; Gibson et al., 1997; De Casson & Laugier, 1999). These systems are limited by the computing hardware, and only perform simple simulations. With the development of computer graphics technology, later methods and products either base the simulation on manually built mesh models and predefined animation or replay real videos captured for VR simulators (Li & Li, 2022; Lesch et al., 2020). They require expensive human labor or advanced hardware or VR video capture but provide limited interaction. Recently, Yang et al. (2024c) proposed SimEndoGS, a data-driven system based on 3DGS and MPM. However, they use an elastic model to capture tissue motion and only support minor interactions like tiny force impulses.

## 3 METHOD

We propose SurgiSim (as illustrated in Figure 1), an automatic system for surgical simulation, which constructs realistic surgical scenes as well as perform accurate simulations. The following sections

detail the methodology: Sec. 3.1 details foundational techniques. Sec. 3.2 describes the reconstruction of a high-quality static simulation environment from a surgical video using a multi-stage optimization process with trajectory and geometric regularization. Sec. 3.3 demonstrates how we accurately model the complex deformations of tissues, thereby facilitating high-quality simulations.

## 3.1 PRELIMINARIES

**3D Gaussian Splatting.** We use 3D Gaussian Splatting (Kerbl et al., 2023), a novel differentiable rendering method which represents scenes with collections of anisotropic 3D Gaussian Kernels $\mathcal{G} = \{G_i : \mu_i, o_i, \Sigma_i, C_i\}_{i=1}^N$, where $\mu_i, o_i, \Sigma_i, C_i$ represents the position, opacity, covariance matrix, and spherical harmonic (SH) coefficients of the $i_{th}$ Gaussian from all $N$ Gaussian kernels. The covariance matrix $\Sigma$ is further decomposed into rotation matrix $R$ and scaling matrix $S$. To render an image through the differentiable rasterization of 3DGS, 3D Gaussian kernels will be projected onto the image plane, and the RGB color is computed as

$$\hat{\mathbf{C}} = \sum_{i \in \{N\}} \alpha_i \, \mathrm{SH}(d_i, C_i) \prod_{j=1}^{i-1} (1 - \alpha_j), \tag{1}$$

where SH denotes the computation of color values based on the given view and SH coefficients, $d_i$ is the view direction from the camera to $G_i$, and $\alpha_i$ represents the effective opacity ordered by z-depth, calculated by multiplying the 2D Gaussian weight with each point's inherent opacity $o_i$. To integrate 3DGS into the simulation pipeline, we view Gaussian kernels as particles carrying properties.

**Continuum Mechanics.** Continuum mechanics models motion with a transformation map $\mathbf{x} = \phi(\mathbf{X}, t)$, where $\mathbf{x}$ represents a material point in the world space $\Omega^t$ at time $t$, deformed from point $\mathbf{X}$ in the undeformed material space $\Omega^0$. The deformation gradient, $\mathbf{F} = \frac{\partial \phi}{\partial \mathbf{X}}$, describes local motion and strain (Bonet & Wood, 1997). In continuum mechanics, the two primary constraints are mass conservation and momentum conservation, given by

$$\int_{\Omega_\epsilon^t} \rho(\mathbf{x}, t) = \int_{\Omega_\epsilon^0} \rho(\mathbf{X}, 0), \; \rho(\mathbf{x}, t)\dot{\mathbf{v}}(\mathbf{x}, t) = \nabla \cdot \boldsymbol{\sigma}(\mathbf{x}, t) + \mathbf{f}^{\text{ext}}, \tag{2}$$

where $\Omega_\epsilon^t \in \Omega^t$ means an infinitesimal region, $\rho$ and $\mathbf{v}$ denote the density and velocity filed respectively, and $\mathbf{f}^{\text{ext}}$ is the external force. $\boldsymbol{\sigma}$ is the Cauchy stress tensor, usually related to a given energy $\Psi$. Details on $\boldsymbol{\sigma}$ will be introduced in Sec. 3.3.

**Marerial Point Method.** Material Point Method (MPM) discretizes the continuum into a collection of Lagrangian particles. The mass conservation of each particle ensures the overall mass conservation. Following Stomakhin et al. (2013), we use particle-to-grid (P2G) and grid-to-particle (G2P) to transfer properties between these particles and an Eulerian grid. The momentum conservation is ensured on the grid where the calculation is simpler and more natural. In each simulation step, the mass and momentum are first transferred from particles onto the grid. The stress tensor is used to update the grid velocities, and the velocities will be transferred back to update particle states. The velocities on the grid are updated as

$$\mathbf{v}_i^{n+1} = \mathbf{v}_i^n - \frac{\Delta t}{m_i} \cdot \mathrm{P2G}(\{\boldsymbol{\sigma}\}), \; \mathbf{v}_j^{n+1} = \mathrm{G2P}(\{\mathbf{v}_i^{n+1}\}) \tag{3}$$

where P2G is the transfer of stress from particles to the grid and G2P is the reverse transfer. $n$ denotes the $n$-th simulation step, each lasts for $\Delta t$, and $\mathbf{v}_i$, $\mathbf{v}_j$ denote velocity vectors on the $i$-th grid node and $j$-th particle respectively. Gravity is ignored in our implementation. Please refer to our appendix for further details on MPM.

## 3.2 SIMULATION ENVIRONMENT SETUP

In this section, we first describe the process of constructing a static simulation environment (canonical 3D scene) from a dynamic surgical video. Subsequently, we detail the regularization techniques designed to enhance geometric consistency, along with the surface thickening methods used to complete the canonical scene.

**Data Preparation.** Given a monocular RGB surgical video $\{\mathbf{C}_o^t\}_{t=1}^T$ with $T$ frames, we first use SAM (Kirillov et al., 2023) segment frames to generate tissue masks $\{\mathbf{M}^t\}_{t=1}^T$. Then we employ a video inpainting method (Li et al., 2022b) to inpaint the RGB frames on the areas covered by surgical instruments with $\{\mathbf{M}^t\}_{t=1}^T$. These inpainted images $\{\mathbf{C}^t\}_{t=1}^T$ are input to Depth Anything v2 model (Yang et al., 2024a) to estimate depth $\{\hat{\mathbf{D}}^t\}_{t=1}^T$.

**Canonical Scene Reconstruction.** The completed frames and their corresponding depths from data preparation are used to build a canonical 3D scene composed of 3D Gaussians, which later serves as a simulation environment. Inspired by Wu et al. (2024); Liu et al. (2024b), we use a deformation field $\mathbf{D}$ to model the 4D deformation of a canonical 3D Gaussian model $\mathcal{G}^0$. To achieve this, we start by initializing a coarse 3D Gaussian model using RGBD projection. This involves using the inpainted first frame and the corresponding estimated depth. Pixels in other frames will be reprojected into the coarse 3D Gaussian model if they belong to the mask area of all previous frames. The deformation field $\mathbf{D}$ is composed of a set of multi-resolution feature planes $\{\mathbf{V}_{ij}\} \subset \mathcal{R}^{h \times lN_i \times lN_j}$ and an MLP $\boldsymbol{\theta}$, where $h$ is the hidden feature size, $l$ is the upsampling scale and $N$ is the resolution parameter. To query the deformation of a Gaussian kernel $G_k : \mu_k, \alpha_k, \Sigma_k, C_k$, we first calculate the voxel feature:

$$f_v = \bigcup_l \text{interp}(\mathbf{V}_{ij}, \mu_k), \ ij \in \{xy, xz, xt, yz, yt, zt\}. \tag{4}$$

Here $\text{interp}$ denotes 4-nearest bilinear interpolation. Then the feature is decoded by the MLP,

$$\Delta\mu, \Delta o, \Delta R, \Delta S = \boldsymbol{\theta}(f_v), \tag{5}$$

and the attributes of the deformed Gaussian can be computed as

$$G_k' = (\mu_k', o_k', \Sigma_k', C_k) = (\mu_k + \Delta\mu, o_k' + \Delta o, \Sigma(R_k + \Delta R, S_k + \Delta S), C_k). \tag{6}$$

Different from previous methods (Wu et al., 2024; Liu et al., 2024b), we allow the opacity to change during deformation. The opacity of tissues would change significantly when we pull it hard, unlike common objects. To render the deformed Gaussian model at a certain timestamp for optimization, we use the deformed attributes calculated above for rasterization in Eq. 1.

**Multi-Stage Optimization with Trajectory and Geometric Regularization.** Our key goal is to create a surgical scene that allows for high-quality surgical simulation rather than generating high-fidelity images at specific timestamps, which all previous tissue reconstruction methods (Yang et al., 2023b;a; Liu et al., 2024b; Xie et al., 2024b) aimed for. Thus, our design focuses on maintaining the geometric consistency of the canonical Gaussian model $\mathcal{G}^0$ during the deformation process. Specifically, we design an explicit trajectory optimization strategy to prevent positional interleaving during deformation, which otherwise leads to faulty optimization of the canonical model. This optimization requires that the trajectories of Gaussian points within any small region tend to be parallel within a short period of time. For a Gaussian model $\mathcal{G}$, we first find the $k$-nearest neighbor point set $N_i$ for each Gaussian $G_i$. For a certain timestamp $t$, the regularization is given by

$$\mathcal{L}_{traj} = \sum_{G_j, G_k \in N_i} \frac{\Delta\boldsymbol{\mu}_j^t \cdot \Delta\boldsymbol{\mu}_k^t}{\|\Delta\boldsymbol{\mu}_j^t\| \cdot \|\Delta\boldsymbol{\mu}_k^t\|} \cdot \|\Delta\boldsymbol{\mu}_j^t\| \cdot \|\Delta\boldsymbol{\mu}_k^t\| = \sum_{G_j, G_k \in N_i} \Delta\boldsymbol{\mu}_j^t \cdot \Delta\boldsymbol{\mu}_k^t. \tag{7}$$

Here $\Delta\boldsymbol{\mu}_i^t$ means the deformation vector from the previous timestamp, i.e., $\mu_i^t - \mu_i^{t-1}$. This regularization consists of two main parts. The first part requires the deformation direction to be parallel to avoid tangential misalignment in the direction of motion. The second part requires the movement length to be small to avoid radial misalignment.

The multi-stage trajectory optimization contains three stages: 1) Optimizing only the canonical model $\mathcal{G}^0$ and the deformation field $\mathbf{D}$ (the feature planes and the MLP) to estimate the coarse trajectory. 2) Adding the trajectory regularization to refine the trajectory of all Gaussian kernels. 3) Freezing the deformation module and only optimizing the attributes of the canonical model $\mathcal{G}^0$.

Additionally, a geometric regularization proposed to prevent excessively large or extremely anisotropic Gaussian kernels is employed across all the stages mentioned above. It is defined as:

$$\mathcal{L}_{geo} = \sum_{i \in \{N\}} \left( \text{ReLU}\left( \max(S_i) - r_m \right) + \text{ReLU}\left( \frac{\max(S_i)}{\min(S_i)} - r_{ani} \right) \right), \tag{8}$$

where $r_m = 1$ is the max scaling limit, and $r_{ani} = 3$ is the anisotropic factor.

---

**Algorithm 1:** Surface Thickening Algorithm

---

**Input** : The canonical Gaussian model $\mathcal{G}^0$
**Output:** A thickened model for simulation $\mathcal{G}_{\mathrm{d}}^0$

1 Initialize $\mathcal{G}_{\mathrm{d}}^0 \leftarrow \mathcal{G}^0$;
2 $\{x_{\max}, x_{\min}, y_{\max}, y_{\min}, z_{\min}, z_{\max}\} \leftarrow$ the tight bounding box of $\mathcal{G}^0$;
3 **for** Layer $l$ in range(1, 1000) **do**
4     **for** Each Gaussian kernel $G_i^0$ in $\mathcal{G}^0$ **do**
5         $G_i^l \leftarrow G_i^0$;
6         The position of $G_i^l$: $\mu_i^l \leftarrow \mu_i^0 \cdot (\mathrm{rand}(3) + l)/1000$;
7         **if** $G_i^l$ in $\{x_{\max}, x_{\min}, y_{\max}, y_{\min}, z_{\min}, (1 + \frac{1}{4})z_{\max}\}$ **then**
8             $\mathcal{G}_{\mathrm{d}}^0 \leftarrow \mathcal{G}_{\mathrm{d}}^0 \cup \{G_i^l\}$ ;
9 **return** $\mathcal{G}_{\mathrm{d}}^0$;

---

**Surface Thickening.** Due to the limited camera view, the reconstructed canonical model $\mathcal{G}^0$ is a single surface with invalid thickness and volume for simulation. To address this, we apply a surface thickening method, pushing the Gaussian kernels in $\mathcal{G}^0$ along the z-axis with a certain probability. The thickening algorithm is described in Algorithm 1.

### 3.3 Accurate and Realistic Simulation

In this section, we first introduce how we model the complex physical effects of tissues with the Visco-Elastic model. Then we demonstrate how we use this model and input video to infer the accurate tissue physical parameters. To better understand our physical model, we employ conventional notations from physics. This may lead to some repetition of the previously defined notations. These corrupted definitions are only used in this section, and their meaning will be re-defined.

**Modeling Tissues with Visco-Elasticity Deformation.** Previous methods (Xie et al., 2024a; Zhang et al., 2024) primarily employ elastic models for simulation, which restricts their performance to jelly-like effects that oscillate back and forth. However, in surgical settings, due to the complex composition of the tissues, tissue deformations are usually very complicated in their physical behaviors, which significantly differ from these jelly-like behaviors. To better model these specific effects, we involve a Visco-Elastic model (Rodríguez et al., 2024), which consists of an elastic part and a viscous part to model the complex deformations.

For the elastic part, we use fixed corotated elasticity. The energy function is defined as:

$$\Psi(\mathbf{F_E}) = \Psi(\mathbf{\Sigma_E}) = \mu_{\mathbf{E}} \sum_i (\sigma_{\mathbf{E},i} - 1)^2 + \frac{\lambda_{\mathbf{E}}}{2}(\det \mathbf{F_E} - 1)^2, \tag{9}$$

where $\mathbf{F_E}$ is decomposed into $\mathbf{U_E}\mathbf{\Sigma_E}\mathbf{V_E}^\top$ through SVD, $\sigma_{\mathbf{E},i}$ are the singular values of $\mathbf{F_E}$. $\mu_{\mathbf{E}} = \frac{E}{2(1+\nu)}$ and $\lambda_{\mathbf{E}} = \frac{E\nu}{(1+\nu)(1-2\nu)}$ are physical parameters, namely Shear modulus and Lamé modulus, computed from the Young's modulus $E$ and Poisson's ratio $\nu$. The Cauchy stress tensor for the elastic part is $\boldsymbol{\sigma_E} = \frac{1}{\det \mathbf{F}} \frac{\partial \Psi}{\partial \mathbf{F_E}}(\mathbf{F_E})\mathbf{F_E}^\top$. The update rule of $\mathbf{F_E}$ is

$$\mathbf{F}_{\mathbf{E},j}^{n+1} = \mathbf{F}_{\mathbf{E},j}^n(\mathbf{I} + \Delta t \cdot \nabla \mathbf{v}_j), \tag{10}$$

and $j$ denotes the $j$-th particle.

As for the viscous part, we get inspiration from Simo & Miehe (1992) and Johnson & Quigley (1992) and propose a simple model regarding viscous energy dissipation to integrate it into the MPM method. The viscous energy dissipation is described through a dissipation potential, given by

$$\Psi(\frac{\partial \mathbf{F_v}}{\partial t}) = \frac{1}{2}\eta_{\mathbf{v}} \mathrm{tr}\left(\left(\frac{\partial \mathbf{F_v}}{\partial t}\right)^\top \frac{\partial \mathbf{F_v}}{\partial t}\right), \tag{11}$$

where $\eta_{\mathbf{v}}$ is the viscosity coefficient. The gradient of $\mathbf{F_v}$ can be updated by $\frac{\partial \mathbf{F_v}}{\partial t} = \gamma \cdot \mathbf{D}$, and $\mathbf{D} = \frac{1}{2}(\nabla \mathbf{v}_j + \nabla \mathbf{v}_j^\top)$ is the symmetric part of the grid velocity. The viscous Cauchy tensor is

Figure 2: Illustration of our physical parameter estimation. SurgiSim automatically infers physical parameters by minimizing the discrepancy between rendered simulation results and the input video through differentiable MPM and rasterization.

$\boldsymbol{\sigma}_{\mathbf{v}} = \det(\mathbf{F}_{\mathbf{v}}) \cdot 2\eta\mathbf{D}$, and $\mathbf{F}_{\mathbf{v}}$ can be updated by

$$\mathbf{F}_{\mathbf{v},j}^{n+1} = \mathbf{F}_{\mathbf{v},j}^{n+1}(\mathbf{I} + \gamma_{\mathbf{v}}\Delta t \cdot \mathbf{D}) \tag{12}$$

The overall stress tensor $\boldsymbol{\sigma} = \boldsymbol{\sigma}_{\mathbf{E}} + \boldsymbol{\sigma}_{\mathbf{v}}$ is used to update the grid velocity.

**Tissue Motion Estimation.** To obtain accurate physical parameters, we involve extracting the physical priors of tissues from surgical videos. This process comprises two main steps: 1) recovering the tissue motion induced by external forces from instruments as captured in the input video, and 2) refining physical parameters by minimizing the discrepancies between the simulated dynamics guided by the tissue motion and input video.

One straightforward way to recover motion in MPM simulation is to estimate the forces involved. However, due to uncertainties in system dynamics, accurately estimating forces is a challenging task (Peng et al., 2019). Instead, we turn to estimating the 3D trajectory of tissues. We start by selecting pixels on the tissues near the contact point with the surgical instruments and employ a 2D dense optical tracking method (Le Moing et al., 2024) to capture the 2D trajectories of tissue movements directly influenced by the external forces. We then augment these 2D trajectories with estimated depth values, $\{\hat{\mathbf{D}}^t\}_{t=1}^T$, to derive 3D trajectories, $\{\mathbf{p}^t\}_{t=1}^T$.

To accurately recover motion in the video, we update the velocity during MPM at time $t$. Specifically, we adjust the grid velocity in a small region $\mathcal{B}^0 \in \Omega^0$ around the starting point of the 3D trajectory $\mathbf{p}^0$. The velocity is updated according to the following formula:

$$\mathbf{v}_{\mathcal{B}^0}^t = \frac{p^{n_t+1} - p^{n_t}}{\Delta T}, \tag{13}$$

where $\Delta T$ is the video frame duration and $n_t$ is the frame index that $n_t\Delta T \le t < (n_t + 1)\Delta T$.

**Physical Parameter Estimation.** We then use the input video to estimate the accurate physical parameters of tissues. This estimation is achieved by minimizing the discrepancies between frames from the input video and the simulation results driven by estimated motion at the time of the frames, as shown in Figure 2.

Firstly, we run $\frac{t}{\Delta t}$ simulation steps to get the simulated model $\mathcal{G}_d^t$ deformed from the dense model $\mathcal{G}_d^0$, and then rasterize the model as $\hat{\mathbf{C}}_s^t = \text{rasterize}(\mathcal{G}_d^t, \text{view})$, where $\text{rasterize}$ is short for Eq. 1. In this way, we align the simulation results with input video. We then optimize the parameters using

$$\mathcal{L}_v = \|\mathbf{C}_o^t - \mathbf{C}_s^t\|_1 \cdot \mathbf{M}^T. \tag{14}$$

Due to the influence of previous simulations on subsequent ones, we implement supervision in a rolling manner. Specifically, we begin by optimizing with the first $k$ video frames. Then, for each new round, we start from the first guide frame and add an additional $k$ frames until the guide length

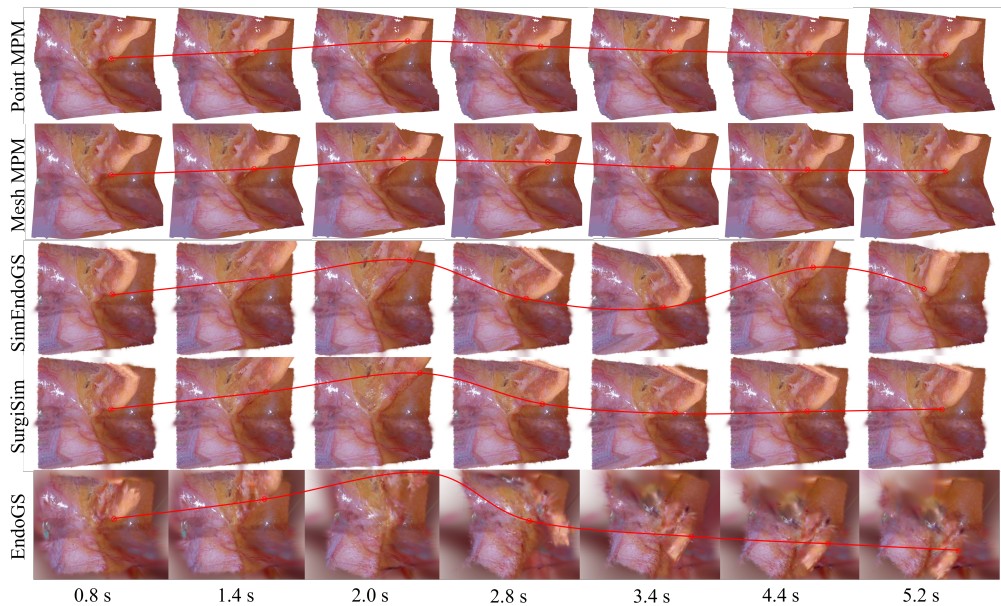

Figure 3: Visualization of simulations. We show the motion of the tissues with a red line. SurgiSim consistently produces the most realistic simulation dynamics. Please refer to the supplementary material for more simulation results.

surpasses the video length. A parameter smoothing regularization is conducted every $k$ frame. For each neighbor group $N_i$ for each Gaussian $G_{d,i}$, we apply a total variation loss:

$$\mathcal{L}_{tv} = \underset{G_{d,j} \in N_i}{\text{MSE}} (\xi_j - \xi_i), \tag{15}$$

where $\xi$ is one of the physical parameters, namely $\mu_{\mathbf{E}}$, $\eta_{\mathbf{v}}$ and $\gamma_{\mathbf{v}}$. We set $\nu_{\mathbf{E}}$ to a constant value of $0.45$, because we have found that variations within the valid range have little impact on the results.

## 4 EXPERIMENTS

### 4.1 IMPLEMENTATION DETAILS.

**Dataset.** We evaluate our method using the EndoNeRF dataset (Wang et al., 2022), which comprises several surgical video clips totaling 807 frames. Each clip, captured by stereo cameras from a single viewpoint, spans 4-8 seconds and shows typical soft tissue scenarios encountered in robotic surgery, including complex non-rigid deformations. We utilize 5 of these clips to establish our simulation environments, excluding one clip that involves significant tissue cutting. Based on these 5 clips, we set up a simulation environment for each and perform a total of 10 typical operations.

**Environment and Simulation Setup.** Our environment setup employs a multi-stage optimization process consisting of 5000 iterations. The initial stage, comprising the first 50% of iterations, focuses on optimizing the canonical model $\mathcal{G}^0$ and the deformation field $\mathbf{D}$. This is followed by a trajectory regularization stage from 50% to 70% of the total iterations. A refinement period then occurs from 70% to 90%, allowing for further optimization of the deformation field after its correction by trajectory optimization. In the final stage, occupying the last 10% of iterations, we freeze the deformation field and fine-tune the canonical model. For the simulation, we adapt the MPM framework from Xie et al. (2024a); Zong et al. (2023). The thickened model $\mathcal{G}^0_{\mathrm{d}}$ is normalized into a 2-unit cube, overlaid with an Eulerian grid of resolution $50 \times 50 \times 50$. To synchronize our physical simulation with real-world time, we established a conversion rate where 10,000 simulation steps correspond to one second of video at 25 fps. For each operation in the simulation, we conduct 80k simulation steps to form an 8-second video. All experiments were conducted on a machine equipped with a Core i7-13700K CPU and a single NVIDIA RTX 4090 GPU, running Ubuntu 24.04. The code will be made public to promote the virtual surgery.

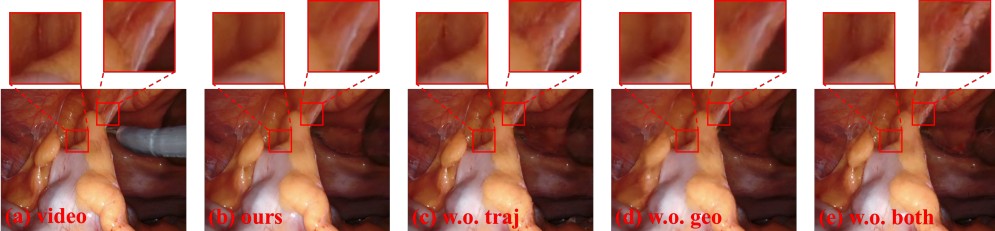

Figure 4: Ablation on the trajectory and geometric regulation. (a) is the input video as a reference. (b) is the result of multi-stage optimization. (c), (d) and (e) are the results of optimization without trajectory regularization, geometric regularization, and both, respectively.

**User Study.** To evaluate the fidelity of our simulations, we conducted a user study involving 68 participants. These participants were categorized into two distinct groups: the Surgeons group, consisting of 44 professional medical personnel, and the Ordinary group, comprising 24 laypersons with no medical background. For More details, please refer to our appendix.

## 4.2 COMPARISONS ON SURGICAL SIMULATIONS

We first provide qualitative comparisons on surgical simulation in Figure 3. We primarily compare with three simulation baselines: 1) SimEndoGS (Yang et al., 2024c), which we implement using PhysGaussian (Xie et al., 2024a) and adapt to our reconstructed simulation environments; 2) A pointcloud-based MPM approach; 3) A triangle mesh simulation with the same MPM. The MPM is built on the string-mass model from Taichi Hu et al. (2019) framework. All methods share the same manipulation during simulation. The simulation results demonstrate that SurgiSimachieves superior simulation results compared to baseline methods in both visual quality and physical accuracy. Our approach generates more realistic tissue deformation responses and interaction dynamics, closely mimicking the behavior observed in actual surgical scenarios. In contrast, SimEndoGS, which relies heavily on manually tuned parameters, often produces unrealistic viscoelastic behavior that deviates from true tissue properties. The baseline MPM, both in their point cloud and mesh-based implementations, suffer from fragmentation, artifacts and unnatural motion during simulation.

Besides, we show Y-T slices across all simulation steps during simulation in Figure 5. Results demonstrate that our simulated tissue rapidly reverts to a quiescent state following a rebound. Conversely, results from SimEndoGS show that the tissue continues to exhibit substantial oscillations post-rebound. This discrepancy is attributed to the proposed Visco-Elastic model which effectively damps out oscillations by simulating the inherent viscoelastic properties of the tissue.

The user study results are shown in Table 1. SurgiSim receives a significant preference across both groups, with a 77.8% preference rate among ordinary viewers and 57.8% among surgeons, significantly outperforming baseline methods. Interestingly, surgeons show a more balanced evaluation, suggesting their professionalism for technical nuances, yet still strongly favored SurgiSim.

| Group | SurgiSim | SimEndoGS | Point MPM | Mesh MPM | SurgiSim | w.o. video guide |
|---|---|---|---|---|---|---|
| Surgeons | **0.578** | 0.256 | 0.100 | 0.067 | **0.656** | 0.344 |
| Ordinary | **0.778** | 0.213 | 0.005 | 0.004 | **0.829** | 0.171 |

Table 1: **User Study** on the simulation results. The values are the mean percentages of each choice.

## 4.3 ABLATION AND ANALYSIS

**Simulation Environment Setup.** We compare simulation results using two different environment setups: one derived from a standard dynamic reconstruction method Liu et al. (2024b) and our geometrically consistent environment. Results are shown in Figure 3. Due to geometric inconsistencies, the baseline environment leads to significant artifacts during simulation. In contrast, our geometrically consistent environment enables stable and realistic tissue deformations.

**Multi-Stage Optimization.** In the multi-stage optimization, we employed trajectory regularization and geometric regularization to enhance the geometric consistency. To assess their individual

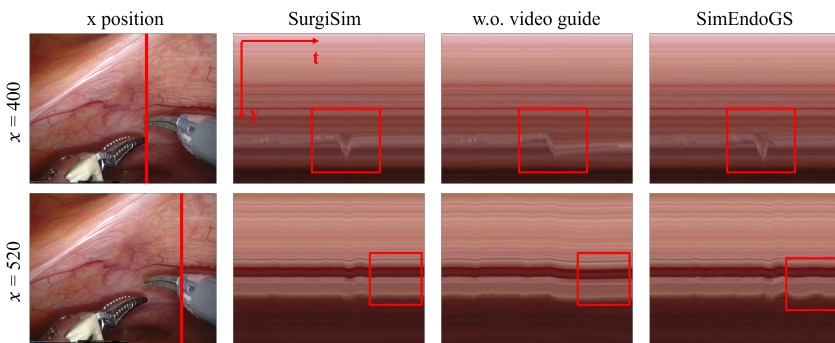

Figure 5: Y-T slices of simulation dynamics. The slices at $x = 400$ capture the motion of tissues being pulled and then rebound. The slices at $x = 520$ capture the oscillations of the tissue after rebound.

contributions, we conduct ablation studies, and results are shown in Figure 4. The result without trajectory regularization (c) achieves overall sound quality, but the surfaces and edges of tissues are rough due to the lack of geometric consistency. The result without geometric regularization (d) contains Gaussian kernels with severe anisotropy, which results in burrs on the surface. These spiked kernels would result in unrealistic protrusions during surgical simulations.

We also provide a quantitative comparison in Table 2. Since the canonical model corresponds to the initial timestamp, rendering metrics are calculated over the first 25 frames. Without geometric regularization, the quality of results is hindered by burrs on the surface. Omitting trajectory regularization doesn't affect the metrics much. As previously discussed, this regularization primarily influences the geometry of the canonical model rather than rendering quality.

| Metric | **SurgiSim** | w.o. traj | w.o. geo | w.o. both |
|---|---|---|---|---|
| PSNR↑ | **37.114** ± 3.7019 | 37.036 ± 4.3512 | 36.582 ± 3.6153 | 36.750 ± 3.6725 |
| SSIM↑ | **0.9772** ± 0.0147 | 0.9770 ± 0.0191 | 0.9730 ± 0.0143 | 0.9754 ± 0.0155 |
| LPIPS↓ | **0.0436** ± 0.0245 | 0.0442 ± 0.0359 | 0.0505 ± 0.0230 | 0.0465 ± 0.0268 |

Table 2: Quantitative results in terms of rendering quality.

**Physical Parameter Estimation.** Figure 5 shows the results of SurgiSim without video-guided parameter estimation. While the tissue can rebound quickly because of good elastic parameters in the result of SurgiSim, the tissue result without estimation would rebound very slowly with severe damping due to faulty viscous parameters. We also report the participants' preferences on the results with and without physical parameter estimation in Table 1. In both groups, participants show an obvious preference for the results with the inferred physical parameters.

## 5 CONCLUSION AND FUTURE WORK

In this paper, we present SurgiSim, an automated and flexible method for transforming monocular surgical videos into simulation-ready scenes, enabling accurate physics surgical simulations within these environments. SurgiSim employs multi-stage optimization with trajectory and anisotropic regularization to construct a geometrically consistent simulation environment. By incorporating a Visco-Elastic deformation model and precise physical parameters extracted from real videos, SurgiSim shows highly realistic and accurate tissue deformations during simulation. We hope that SurgiSim could contribute to the development of more diverse and realistic surgical simulations.

**Limitations.** In the simulation environment setup, our method could not handle the invisible part on the side of the tissues, which caused flaws in the texture formed by our thickening method. As for simulation, firstly, the physical prior in the videos is sometimes not enough for only part of the tissues are actually moving and contribute to the parameter estimation. Secondly, our method does not yet support more complicated operations like cutting and topological inversions of structure.

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

## A  APPENDIX

### A.1  MATERIAL POINT METHOD

The full MPM methods include particle-to-grid (P2G) and grid-to-particle (G2P) to transfer properties between these particles and an Eulerian grid. Following Stomakhin et al. (2013); Xie et al. (2024a), we use $C^1$ continuous B-spline kernels for two-way transfer. The mass and momentum are transferred from particles to grid nodes:

$$m_i = \sum_p m_p \, w_{ip}, \tag{16}$$

$$m_i \mathbf{v}_i = \sum_p m_p \left( \mathbf{v}_p + \mathbf{C}_p(\mathbf{x}_i - \mathbf{x}_p) \right) w_{ip}, \tag{17}$$

where $m_i$ is the mass at grid node $i$, $m_p \mathbf{v}_p$ are the mass and velocity of particle $p$ and $\mathbf{v}_i$ is the velocity at grid node $i$. $\mathbf{x}_i$ and $\mathbf{x}_p$ are the positions of grid node $i$ and particle $p$, respectively, $w_{ip}$ is the B-spline weighting function between particle $p$ and grid node $i$ and $\mathbf{C}_p$ is the affine velocity matrix of particle $p$, capturing local velocity gradients.

After grid velocities are updated, particle velocities and affine matrices are interpolated from the grid:

$$\mathbf{v}_p^{n+1} = \sum_i w_{ip} \mathbf{v}_i^{n+1}, \tag{18}$$

$$\mathbf{C}_p^{n+1} = \frac{12}{\Delta x^2 (b+1)} \sum_i w_{ip} \mathbf{v}_i^{n+1} (\mathbf{x}_i - \mathbf{x}_p)^\top, \tag{19}$$

where $\mathbf{v}_p^{n+1}$ is the updated velocity of particle $p$, $\mathbf{v}_i^{n+1}$ is the updated velocity at grid node $i$, $\mathbf{C}_p^{n+1}$ is the updated affine velocity matrix for particle $p$, $\Delta x$ is the grid spacing and The term $(\mathbf{x}_i - \mathbf{x}_p)^T$ represents the transpose of the position difference vector.

## A.2   ADDITIONAL COMPARISON

Table 3 shows the reconstruction quality of our reconstruction with multi-stage optimization, compared with EndoNeRF (Wang et al., 2022), EndoSurf (Zha et al., 2023), LerPlane (Yang et al., 2023b) and 4D-GS (Wu et al., 2024). We follow the evaluation setting of (Xu et al., 2024). While our reconstruction method is not designed for novel-timestamp synthesis ability which the evaluation metrics focus on, it still achieves relatively good results.

| Metric | SurgiSim | EndoNeRF | EndoSurf | LerPlane | 4D-GS |
|--------|----------|----------|----------|----------|-------|
| PSNR↑ | 34.605 | 27.077 | **34.795** | 34.643 | 22.832 |
| SSIM↑ | **0.960** | 0.900 | 0.945 | 0.922 | 0.827 |
| LPIPS↓ | 0.076 | 0.107 | 0.119 | **0.072** | 0.368 |

Table 3: Full reconstruction comparison of SurgiSim and other methods.

## A.3   DETAILS ON USER STUDY

The study was structured into two parts. In the first part, participants were tasked with selecting the most accurate simulation from four options. In the second part, they were required to choose the superior simulation between the two presented results. Each participant was exposed to nine sets of simulation results, with the order of the sets randomized to prevent order bias. Before analysis, we executed a basic data-cleaning process to remove any invalid or outlier responses.

