# OpenReview forum: "Realistic Surgical Simulation from Monocular Videos"
_ICLR.cc/2025/Conference — ICLR 2025 Conference Withdrawn Submission_

### Official Review · Reviewer_Mqpg · 2024-10-27

**Soundness:** 2
**Presentation:** 2
**Contribution:** 2
**Rating:** 3
**Confidence:** 3

**Summary:**

The paper present a method to simulate realistic surgical scene. The core method uses a physics-based Maxwell model to restore the complex deformations of tissues. The proposed system aims to improve the surgical training, planning, and robotic surgery systems by offering accurate surgery scene simulation.

**Strengths:**

1. The user study part is good and ensures the usability of the proposed system.

**Weaknesses:**

1. The methodology presented in this paper similar to the paper"SimEndoGS: Efficient Data-driven Scene Simulation using Robotic Surgery Videos via Physics-embedded 3D Gaussians" with no substantial novel improvements. The evaluation in Figure 3 does not suggest advancement in this work when comparing with the SimEndoGS.

2. Lacks comprehensive quantitative evaluation of the proposed method itself, e.g. did not compare with SimEndoGS.

3. Lacks significant testing.

**Questions:**

1. One major confusing point is: this paper states to simulate the dynamic surgery from monocular videos; however, why are the authors use stereo surgery videos, such as EndoNerf, to evaluate the proposed method's performance?

2. Why put the quantitative evaluations with EndoNerf in the Appendix? The core evaluation results should be placed in the main paper.

**Details Of Ethics Concerns:**

Does not provide the IRB approval statement as the paper involves human subjects in user study.

---

### Official Review · Reviewer_oHUM · 2024-10-28

**Soundness:** 3
**Presentation:** 2
**Contribution:** 2
**Rating:** 5
**Confidence:** 4

**Summary:**

This work proposes a framework to better incorporate physics-based deformation in surgical video reconstruction. Existing research in this area often fails to achieve geometrically consistent reconstructions and to simulate complex tissue deformations. To address these challenges, this paper introduces a Visco-Elastic deformation model in 3D Gaussian Splatting (3DGS)-based surgical scene reconstruction.

**Strengths:**

- The paper has a clear motivation. Maintaining a physics-based model for tissue deformation is critical in surgical environments.
- The use of physical parameters helps to maintain the optimization target with clear global information, rather than merely relying on Gaussian parameters. This approach leads to practical significance and explainability.

**Weaknesses:**

- The technical aspects are somewhat unclear. My understanding is that [1] attempts to learn physical properties from prior knowledge, which is leveraged from video diffusion models. Although the video diffusion model (VDM) is an imperfect prior model, it can provide additional information about the material. However, endoscopic reconstruction is primarily a task of reconstruction, where most information is already present in the video. This work does not introduce additional priors apart from the depth estimation model, making it unclear how additional physical information is learned. For example, if the reconstruction results of dynamic scenes resemble the video with great 3D visualization, it suggests similar material properties.
- Can the learned physical properties demonstrate downstream evaluation? Since the evaluation is the same as the earlier works in this field, it is unclear whether the learned physical information provides any benefit.
- The paper describes the work as realistic surgical simulation, which seems to be an overclaim. In fact, this work focuses on surgical reconstruction, an important topic, but it is not accurate to describe it as surgical simulation.


Other minor issues
- The paper is missing a citation for the first work [2] on surgical reconstruction with Gaussian splatting. From Figure 3, this work seems to have already been compared.
- The definitions of the elasticity and visco components, F_E and F_V, are missing. Their meanings should be clarified in Section 3.3.
- In Section 3.2, the paper mentions that the SAM, inpainting model, and Depth Anything are used for the tool masks and depth maps. Since many of the compared methods use the given maps in EndoNeRF, it would be better to compare the results with baselines using the same evaluation setting, i.e., the same tool maps and depth maps. Alternatively, the comparison methods may have already been evaluated in this setting.

[1] Liu, Fangfu, et al. "Physics3D: Learning Physical Properties of 3D Gaussians via Video Diffusion." arXiv preprint arXiv:2406.04338 (2024).

[2] Zhu, Lingting, et al. "Deformable endoscopic tissues reconstruction with gaussian splatting." arXiv preprint arXiv:2401.11535 (2024).

**Questions:**

See Weaknesses.

---

### Official Review · Reviewer_mCUt · 2024-11-01

**Soundness:** 2
**Presentation:** 3
**Contribution:** 3
**Rating:** 5
**Confidence:** 4

**Summary:**

The authors introduce SurgiSim, a method which, given a monocular surgical video (with a
static camera and limited tissue cutting), can create realistic simulations of deformations to
Gaussian Splatting reconstructions of soft tissues from the videos. The work is interesting
and relevant to the ICLR community, given the novelties to trajectory and geometry
optimisation/regularisation, as well as in estimating physical parameters of tissues from
videos.

**Strengths:**

- Interesting approach including a novel multi-stage optimisation of trajectory and geometric regularisation
 - Estimating physical parameters of tissues directly from videos is a welcome  innovation
- Method uses off-the-shelf methods monocular methods, meaning that stereo information and manual segmentations are not required
- Authors perform a user study with 44 surgeons and 24 laypersons, where a  majority find their method to produce the most realistic simulations
- SurgiSim achieves a good reconstruction quality (in terms of PSNR etc.) despite being designed for high-quality simulations (as in Table 3 of the appendix).

**Weaknesses:**

- Limited evaluation, in particular no quantitative results (see questions)
- Details regarding methods and evaluation are missing (see questions)
- Paper is on the whole well-written, but could be clearer and more descriptive in places

**Questions:**

- The authors claim that their system produces “ accurate, high-quality surgical simulation”. However, given that there is no quantitative evaluation of accuracy with respect to ground truth, it cannot be said that the results are accurate. On the other hand, the results are high-quality as indicated by Table 2 and Table 3 (appendix), and as preferred by the user study. Similarly, in appendix A.3 ‘Details on User Study’, it was reported that the participants were tasked with selecting the “most accurate simulation from four options”. Again, accurate is not appropriate word choice - ‘realistic’ would be preferable without experiments demonstrating accuracy.

- Figure 3 appears ‘stitched’ together: It looks like the outputs from different timestamps are separately placed together, and not all well-aligned. Considering this (and despite the appearance being a bit messy), it is not clear how the red line representing tissue motion is produced, and whether it is reliable or how useful it is.

- The authors mention that they set up a simulation environment for 4 different clips of the EndoNeRF dataset, and perform 10 “typical operations”. This lacks detail, as it is not clear what is meant by ‘typical’ (are these patches or points or lines?). How large or varied are these inputs? And how those typical operations were devised - limiting reproducibility and understanding for the reader. This also makes Figures 3 and 5 less meaningful, as it is not clear what should be expected: the qualitative evaluation outcome is highly dependent on knowing the initial conditions and the applied forces/motion, but these are not adequately described. Not to labour this point, but seeing as SimEndoGS was mentioned as only supporting ‘minor interactions like tiny force impulses’, it would be good to know the characteristics of those used in this study.

- The authors mention that ‘only part of the tissues are moving and contribute to the parameter estimation’. Due to this, it may be the case that the simulations appear more or less realistic depending on where in the reconstructed scene forces/motions are applied. For this reason, it would be useful to know under what conditions the proposed method performs well - i.e., how does the realism of the simulations
change depending on how far away from observed moved tissues forces/deformations are applied? It may be the case that simulations with
heuristically set parameters are more reliable in such scenarios.

- The nature of the deformation for Figure 5 is mentioned in passing, however its extent is not - this, together with the fact that the time over which the slices of Figure 5 are depicted make it difficult to say for the reader whether the oscillations are desirable or not (because oscillations of soft tissue can occur). The size, direction and position of the applied force/motion/deformation should be indicated somehow -
oscillations seem only to be behind the instrument in the image, but are not present elsewhere.

- How were the simulations presented to the participants during the user study? Specifically, was the applied force/motion shown to the participants during the user study, such that they could distinguish the applied force/motion from the resulting simulated impact on the surrounding tissues?

- Why was EndoGS included in Figure 3 but omitted from Appendix A.2 ‘Additional Comparison’?

- One thing that might enhance this submission and show that the parameters learned for tissues are ‘accurate’ is to find short videos where the same surgical scene is subjected to one or more separate/distinct deformations.
Here, one of these deformations could be used to estimate the tissue parameters with the proposed method, and using the proposed method of obtaining tissue motions (optical tracking + depth) for the other deformations, the obtained tissue parameters could be fixed while simulations for the other deformations are performed. The simulation outputs might then be comparable to ground truth and a
notion of accuracy could be introduced.

---

### Official Review · Reviewer_mHQD · 2024-11-09

**Soundness:** 3
**Presentation:** 4
**Contribution:** 4
**Rating:** 6
**Confidence:** 4

**Summary:**

This work presents a NeRF-based method to produce a realistic surgical simulation system from surgical videos. The authors combine several methods from computer vision to create a system that maintains physical fidelity. The authors use Gaussian splatting for the scene reconstruction and the material point method to model elastic tissue deformation. They employ additional geometric regularization to maintain consistency across video frames. They develop the simulation environment on 5 video clips. The authors perform a comprehensive user study and demonstrate strong preference to their method.

**Strengths:**

The work proposes a novel solution to alleviate difficulties in surgical simulation pipelines. The current approach use mesh-based tissue modeling, which although are realistic, require manual modeling. The authors combine several existing technologies in a novel way to construct a realistic environment that maintains physical realism. The paper is well written and technically sound.

The user study is another strength of this approach as it is both quantitatively validated and qualitatively with the correct end users in mind.

**Weaknesses:**

The main weaknesses I see in this work are related to generalization, proper application, and certain modeling decisions.
- In the design of regularization for geometry, how do the authors guarantee that no tissue inversions would occur? As described on Line 244 (Multi-Stage Optimization with Trajectory and Geometric Regularization), the authors add soft regularization to prevent large deformation of the Gaussian points. However, this is a soft loss, and I am not sure how well motivated it is. The deformation literature has extensive work on preventing inversions [1,2]. Can the authors comment on how this would apply in the Gaussian splatting case?
- Can the authors comment on the generalizability of their method? As far as I understand, they took video frames from the same type of surgery and tissue, and only used short clips (4-5 seconds). How realistic is this to create a surgical simulation environment?
- Further, the training setup is quite complicated, involgving splitting the training into stages. How was this setup determined, and would it generalize to other types of simulation?
- What is the computational cost of training these models? Would each new simulation require re-training?
- Why were there no quantitative evaluations performed with baselines? Can the authors comment on the difficulty of performing this experiment?


References

[1]G. Irving et al., "Invertible Finite Elements For Robust Simulation of Large Deformation", SIGGRAPH Symposium on Computer Animation (2004)

[2]Smith et al., "Analytic eigensystems for isotropic distortion energies.", ACM TOG, 2019

**Questions:**

see above

---

### Note · Authors · 2024-11-13

I have read and agree with the venue's withdrawal policy on behalf of myself and my co-authors.